# Granuloma and persistent detection of wild-type rubella virus in an immunocompromised patient

Charlotte Pronier,[1] Steven Roger,[1] Juliette Besombes,[1] Francisco Llamas Guttierez,[2] Matthieu Revest,[3] Judith M. Hübschen,[4] Antoinette Perlat,[5] Christelle Vauloup-Fellous,[6,7] Vincent Thibault[1]

**ABSTRACT** We report a case of wild-type rubella virus (genotype 2B) granuloma in a 29-year-old immunocompromised patient with a full vaccination scheme. He had been followed since his early teens for an unlabeled systemic inflammatory disease. On an inguinal node biopsy, histological analysis revealed a nguudiffuse non-necrotic granulomatous inflammation with no identified conventional infectious agent involved in granuloma formation. Further virological investigation revealed rubella virus (RuV) RNA detected by reverse transcription polymerase chain reaction (RT-PCR). Wild-type viral RNA persistence was documented in plasma samples collected up to 25 months after initial detection and also in different sequential biological samples (cerebrospinal fluid and nasopharyngeal swab). In the context of incomplete rubella eradication, RuV must be added to the list of potential triggers of chronic granuloma in immunodeficient patients.

**IMPORTANCE** Rubella has become rare but has not yet disappeared. Rubella virus (RuV) must be added to the list of potential triggers of granuloma, particularly in immunodeficient patients. Diagnosis of RuV granulomas requires molecular testing of biopsy. RuV sequencing distinguishes between wild-type and vaccine-derived viruses and enables epidemiological surveillance.

**KEYWORDS** granuloma, immunodeficiency, rubella virus

A 29-year-old man, born in 1991, had been regularly followed since 2004 for a systemic inflammatory disease of unknown etiology. Hypogammaglobulinemia and lymphopenia were found secondary to immunosuppressive treatment or underlying immune deficiency. In March 2020, an inguinal node biopsy was performed due to the appearance of polyadenopathies and splenomegaly. Histological analysis revealed a diffuse non-necrotic granulomatous inflammation. The residual lymphoid tissue contained no B cells but small T cells. Microbiological testing of the lymph node was negative for *Mycobacterium tuberculosis*, *Bartonella spp.*, and *Leishmania sp*. Given these initial negative results for the infectious agents classically found in granulomas, rubella testing was conducted. Rubella virus (RuV) RNA was detected by RT-PCR. RuV genotype 2B was identified by sequencing.

Available samples were retrospectively analyzed to further document the case (Table 1). While the RuV RT-PCR was negative in plasma drawn 2 weeks prior to the biopsy being performed, we documented the persistence of RuV-RNA in consecutive plasma samples by RT-PCR (ct range from 31 to 36), collected up to 2 years after first detection. RuV-RNA was also detected in different specimens such as cerebrospinal fluid (CSF) and a nasopharyngeal swab, 1 year after its initial detection in the lymph node (Table 1). Regarding the serological profile, the immunoblot showed the presence of anti-E1 protective antibodies, while RuV IgG commercial immunoassays provided negative

**Peer Reviewer** Ludmila Perelygina, Centers for Disease Control and Prevention, Atlanta, Georgia, USA

Address correspondence to Charlotte Pronier, charlotte.pronier@univ-rennes.fr.

The authors declare no conflict of interest.

*[This article was published on 31 January 2025 missing the Data Availability paragraph. The Data Availability was corrected in the current version, posted on 5 June 2025.]*

**TABLE 1** Laboratory investigations for *Rubivirus rubellae* (RuV)[a]

| Sample date (time since RuV RNA detection in lymph node) | (D-15) | (D0) | (M9) | (M10) | (M13) | (M13) | (M13) | (M15) | (M18) | (M20) | (M22) | (M25) |
|---|---|---|---|---|---|---|---|---|---|---|---|---|
| **RuV RT-PCR** | | | | | | | | | | | | |
| **Biological samples** | | | | | | | | | | | | |
| - Plasma | Negative | | | | | Positive (Ct = 33)∞ | Positive (Ct = 33) | Positive (Ct = 33)∞ | | Positive (Ct = 36) | Positive (Ct = 31) | Positive *∞ |
| - Lymph node biopsy* | | Positive (Ct = 29) Genotype 2B*∞ | | | | | | | | | | |
| - CSF | | | Positive (Ct = 33)*∞ | | Positive (Ct = 33)∞ | Positive (Ct = 31) | | | | | | |
| - Nasopharyngeal swab | | | | | | Positive (Ct = 31) | | | | | | |
| - Urine | | | | | | | | Negative | | | | |
| **RuV serology** | | | | | | | | | | | | |
| Anti-RuV IgG (IU/mL, positive if ≥10 IU/mL) | | | Negative* 3.14 | Negative 0.96 | | | Negative 0.38 | | Positive# (35.84) | Negative 8.40 | | |
| Anti-RuV IgM (index)* positive if >1.2 | | | Negative 0.23 | Positive (14.38) | | | | | Equivocal (0.91) | | | |
| Immunoblot* | | | Positive Anti-E1 | Positive Anti-E1 | | | | | | | | |
| Intrathecal IgG synthesis (CSF/serum)* | | | Positive | | | Not tested | | | | | | |

[a]D0 corresponds to the first RuV identification. M corresponds to the number of months since D0. RuV IgG titers were measured by Enzyme-Linked Immuno-Sorbent Assay (ELISA) with Elecsys RuV IgG test on Cobas 6000 and expressed as international units per milliliter. RuV RNA detection was performed by an in-house RT-PCR described by Okamoto et al. (Ct = Cycle threshold). "#" indicates a serum collected 1 month after immunoglobulin administration. "*" highlights analyses performed by the French National Reference Laboratory for Maternofetal Rubella Infections, which used the Enzygnost anti-rubella virus IgG assay (Siemens Healthcare, Marburg, Germany) and the Vidas Rubella IgM assay (bioMérieux, France). Immunoblot was performed with recomBlot rubella IgG (Mikrogen GmbH, Neuried, Germany). Intrathecal synthesis: IgG antibody titers were determined using Enzygnost specific IgG enzyme-linked immunosorbent assay kits (Siemens, Marburg, Germany). CSF and serum were paired for analysis in the same analytical run. "∞" indicates samples used for genotyping.

results on serial samples, except for a serum taken 1 month after the last administration of intravenous immunoglobulin (IVIg). RuV IgM was detected in a single serum taken 10 months after detection of the viral genome in the lymph node. In December 2020, the patient was admitted to the hospital in the presence of neurological symptoms consistent with sensorimotor neuropathy of the lower limbs. These symptoms evolved and led to total hearing loss and tetraparesis in 2022. In this context, two lumbar punctures have been performed at different times from the first viral detection in the lymph node, at M9 (when the neurological signs started) and at M13 (when the motor deficits worsened), respectively. At M13, cerebral Magnetic Resonance Imaging (MRI) showed discrete impairment of cerebral trophicity and hypersignals in the deep white matter, particularly in bilateral parietal periventricular areas, without clear etiological orientation. CSF analyses revealed the presence of RuV RNA at M9 and M13 and specific intrathecal anti-RuV IgG synthesis at M9 (Table 1), while herpes simplex virus and varicella zoster virus (VZV) intrathecal syntheses were negative. Phylogenetic analysis of the RuV sequences obtained at different time points indicated that they clustered separately from previously published genotype 2B sequences from France. The closest relatives identified by Nucleotide BLAST were sequences obtained in 2012 in China and Great Britain, which formed, however, a separate cluster (Fig. 1).

The patient's vaccination record documented a full vaccination scheme with two doses received at the age of 1 and 11 years. His mother declared a normal pregnancy. The mother's immune status to rubella was unknown, as rubella screening during pregnancy has only been mandatory in France since 1992.

Whole-genome sequencing revealed several variants of uncertain significance in the LRBA and DCLRE1C genes in February 2021. No causal anomalies were identified. The patient was rejected for hematopoietic stem-cell transplantation due to his general condition. He was hospitalized for deterioration and died in June 2022.

We describe the detection of a wild-type RuV strain in granuloma from an immuno-compromised patient with a full vaccination scheme. Nearly a hundred cases of RuV granuloma have been reported in the literature (1). Most cases involve vaccine-derived RuV (VDRV) in immunocompromised patients. Only four cases have been reported with wild-type RuV persistence within cutaneous granulomas, all in men (2–5). Two cases occurred in unvaccinated adults (a 77-year-old man with common variable immunodeficiency and a 56-year-old man, with low CD8+ T cell counts). In both cases, a distinct genotype missed by rubella surveillance was amplified from skin biopsies (multiple body sites for the first patient and the left arm for the second). The third case was an 11-year-old child with ataxia telangiectasia and a right leg granuloma, without documented genotype (4). This patient was immunized with two doses at age 9. The fourth case was a 70-year-old man with unknown vaccine status. Cutaneous lesions of the left leg slowly evolved for over 30 years in the absence of immunodeficiency. Wild-type RuV genotype 1a was identified from the skin granuloma during his care for lymphocytic lymphoma (5). No specific features were found in comparison to VDRV-associated granuloma. The inguinal location of the granuloma in our case is uncommon with most granulomas described on the arms, a common site for administering vaccines. The patient, with no travel history, harbored genotype 2B RuV, widely distributed. In France, only two genotype 2B strains were identified in 2004 and 2009, and none of them was similar to our case's strains (6). Interestingly, the sequences of our case obtained at different time points and from different specimen types and locations varied by up to 22 nucleotides within the 739 nucleotide region analyzed. As reported previously for a skin biopsy and a nasopharyngeal swab from a granuloma patient with primary immune deficiency and immunodeficiency-related vaccine-derived rubella viruses from Louisiana (7), we obtained different sequences derived from different clinical specimens collected at the same time (plasma 1 and CSF 2, Fig. 1). In addition, the sequences from the same specimen type differed between the different collection time points and interestingly also between two plasma samples collected at the same time point (plasmas 2 and 3, Fig. 1). These findings corroborate the hypothesis of continuous and independent virus

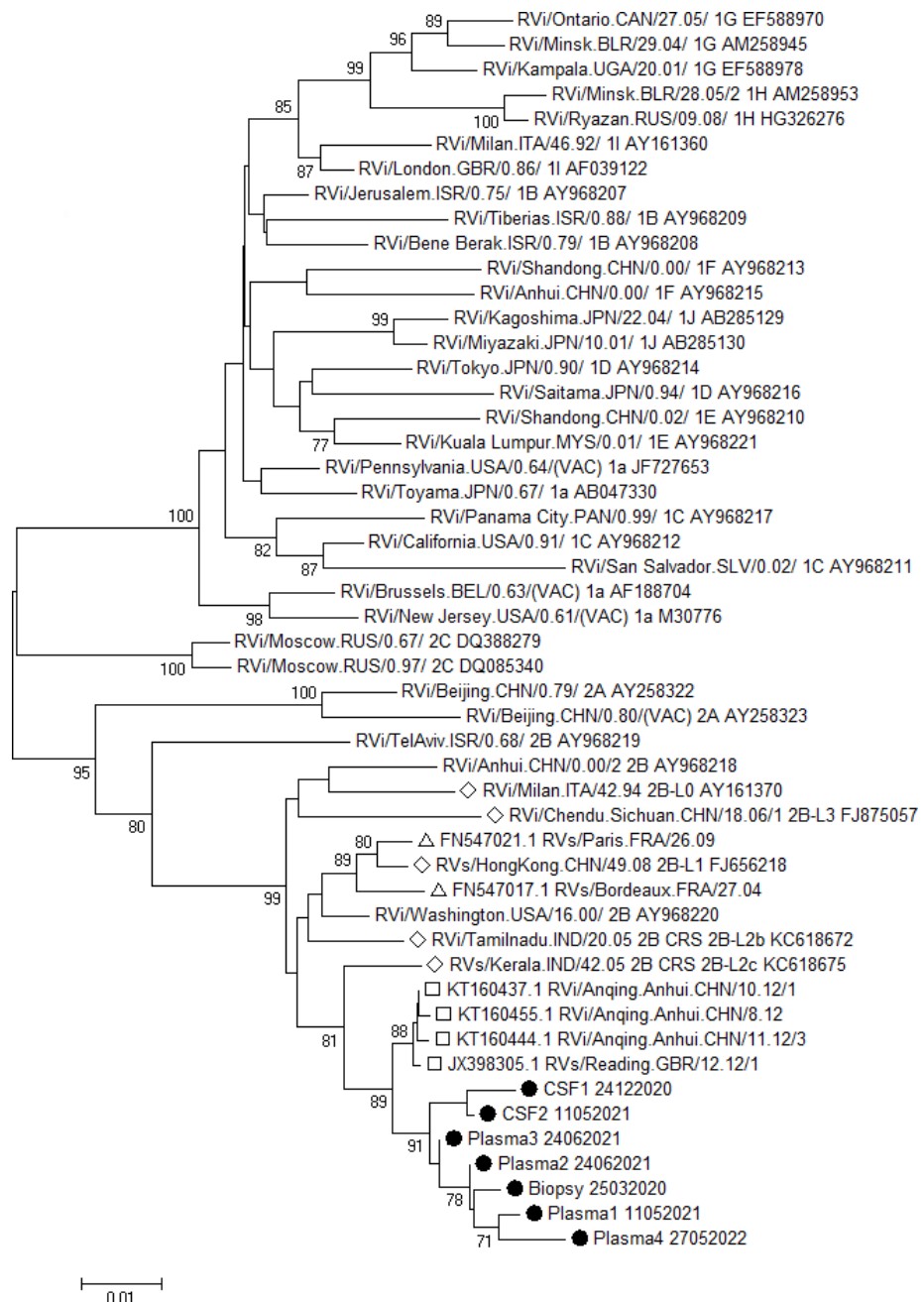

**FIG 1** Phylogenetic trees based on 739 nucleotides of the *Rubivirus rubellae* E1 gene. Phylogenetic analysis was performed using the Neighbor-Joining method and Kimura 2-parameter model and included the World Health Organization (WHO) reference sequences. Bootstrap values of at least 70 (*n* = 1,000 replicates) are shown close to the nodes. The sequences obtained from the patient described here are marked with black dots, the BLAST hits with the highest score with open squares, previously reported 2B sequences from France with open triangles, and 2B lineage references (10) with open diamonds. Please note that the WHO reference sequence RVi/Anhui.CHN/0.00/2_2B_AY968218 serves as a reference for lineage 2B-L4, and the 2B-L2a lineage reference RVi/Seattle.WA.USA/16.00/2B_2B-L2a_JN635293 has the same sequence as the WHO reference strain RVi/Washington.USA/16.00/_2B_AY968220 and is therefore not shown.

evolution at different body sites and a complex virus population structure in chronically infected patients (7), even if infected with a wild-type-derived rubella virus.

In our case, wild-type RuV detection in different sequential samples (biopsy, plasma, CSF, and respiratory samples) was unexpected. While VDRV RNA detection in sequential

samples (respiratory, CSF, and urine) collected over 15 months has been reported in one patient (8), we document wild-type viral RNA persistence in plasma samples collected up to 25 months after its initial detection. RT-PCR in the sole tested urine sample was negative. Three main hypotheses exist concerning the source of RuV in our patient: congenital rubella infection, rubella vaccination failure, or infection before immunization. First, the clinical ocular and cardiac manifestations mentioned in the patient's history are not characteristic for congenital rubella but may rather be related to the patient's immune deficiency. Moreover, no relation between congenital rubella and granuloma has been proven so far, and there is no description of such prolonged persistence of the virus. Second, rubella vaccination may have failed because of undiagnosed immune deficiency at the time of vaccination. It is unlikely as this vaccine is highly immunogenic, and the patient had received two doses. In addition, antibodies against VZV, diphtheria, tetanus, poliovirus, and SARS-CoV-2 were detected, suggesting that the patient was able to mount a humoral immune response. Third, the patient might have been infected before immunization with viral persistence in the immunodeficiency disorder context. In the early 1990s, when the patient was a child, first-dose rubella vaccine coverage was only around 70%, and RuV was still circulating in France. It is thus possible that he was infected before receiving his first dose of rubella vaccine at the age of one, or before the second dose at 11 years. Although phylogenetic analysis suggested no close relation of the sequence to the two genotype 2B strains previously detected in France, there was no travel or exposure history and no symptoms consistent with rubella at the time, which could clarify the source of infection.

No anti-RuV IgG antibodies were detected by commercial assays at different times, except 26 days after immunoglobulin administration. This serological profile might be explained by the patient's immunocompromised status or by a vaccine failure. However, the immunoblot assay detected anti-E1 protective antibodies in samples, prior to any administration of IVIg but after RuV RNA detection in lymph node tissue. Anti-E1 positive results indicate humoral immunity with very good specificity and sensitivity, often outreaching commercial immunoassays (9). Thus, the detection of anti-E1 antibodies reflects a certain immunity acquired either after vaccination or natural infection, which was, however, insufficient to reach viral clearance in this particular context of immuno-suppression. The detection of anti-rubella IgM was surprising. Perelygina et al. reported detection of IgM antibodies to RuV in the sera of two immunodeficient patients and proposed that it may be a marker of virus persistence, potentially useful for identifying patients with infectious immunodeficiency-related VDRV before development of lesions (7). In our case, IgM antibodies to RuV were only detected once, in January 2021, before any IVIg administration, but not in a sample taken one month earlier. IgM detection can be explained either as a result of the chronic infection or by stimulation of the immune system at that time. Of note, we cannot rule out a false IgM reactivity of the positive sample, as often seen with IgM detection for rubella and other infections.

With rubella elimination not yet achieved worldwide, RuV must be added to the list of potential triggers of chronic granuloma in immunodeficient patients. The impact of the long-term shedding of wild-type RuV on rubella eradication efforts is currently difficult to estimate. Positive cultures of both VDRV and wild-type RuV from granuloma patients raise the question of the transmissibility of these viruses in long-term shedding patients (3, 7) and thus the risk of this virus reservoir for elimination efforts.

## ACKNOWLEDGMENTS

Aurélie Sausy and Emilie Charpentier helped with lab investigation and analysis. Dr. Marine de Carlan cared for the patient and helped collect data.

## AUTHOR AFFILIATIONS

[1]Department of Virology, Pontchaillou University Hospital, Rennes, France
[2]Department of Pathology, Pontchaillou University Hospital, Rennes, France

[3]Department of Infectious Diseases and Medical Intensive Care Unit, Pontchaillou University Hospital, Rennes, France

[4]Department of Infection and Immunity, Luxembourg Institute of Health, Esch-sur-Alzette, Luxembourg

[5]Department of Internal Medicine, Pontchaillou University Hospital, Rennes, France

[6]Department of Virology, Rubella National Reference Laboratory, Paris Saclay University Hospital, APHP, Paris, France

[7]Université Paris-Saclay, INSERM U1184, CEA, Center for Immunology of Viral, Auto-immune, Hematological and Bacterial Diseases (IMVA-HB/IDMIT), Fontenay-aux-Roses, France

## PRESENT ADDRESS

Steven Roger, Laboratoire Cerballiance, Pleumelec, France

## AUTHOR ORCIDs

Charlotte Pronier  http://orcid.org/0000-0001-7873-2752
Vincent Thibault  http://orcid.org/0000-0001-6517-2901

## AUTHOR CONTRIBUTIONS

Charlotte Pronier, Conceptualization, Data curation, Formal analysis, Investigation, Supervision, Validation, Writing – original draft | Steven Roger, Data curation, Writing – original draft | Juliette Besombes, Data curation, Writing – review and editing | Francisco Llamas Guttierez, Data curation, Writing – review and editing | Matthieu Revest, Validation | Judith M. Hübschen, Data curation, Investigation, Validation, Writing – review and editing | Antoinette Perlat, Data curation, Investigation, Validation | Christelle Vauloup-Fellous, Data curation, Investigation, Writing – review and editing | Vincent Thibault, Supervision, Writing – review and editing

## DATA AVAILABILITY

Sequence data were submitted to GenBank and are available under accession numbers PV035058 (https://www.ncbi.nlm.nih.gov/nuccore/PV035058.1), PV035059 (https://www.ncbi.nlm.nih.gov/nuccore/PV035059.1), PV035060 (https://www.ncbi.nlm.nih.gov/nuccore/PV035060.1), PV035061 (https://www.ncbi.nlm.nih.gov/nuccore/PV035061.1), PV035062 (https://www.ncbi.nlm.nih.gov/nuccore/PV035062.1), PV035063 (https://www.ncbi.nlm.nih.gov/nuccore/PV035063.1) and PV035064 (https://www.ncbi.nlm.nih.gov/nuccore/PV035064.1).

## ETHICS APPROVAL

The patient's written consent was obtained, and design of the work has been approved by local ethical committees (notice no. 23.122).

## ADDITIONAL FILES

The following material is available online.

Open Peer Review

**PEER REVIEW HISTORY (review-history.pdf).** An accounting of the reviewer comments and feedback.

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
