## [Reviewer comments · Microbiology Spectrum]

Microbiology Spectrum

Granuloma and persistent detection of wild-type rubella virus in an immunocompromised patient.

Charlotte PRONIER, Steven Roger, Juliette Besombes, Francisco Llamas Gutierrez, Matthieu Revest, Judith Hübschen, Antoinette Perlat, Christelle Vauloup-Fellous, and Vincent Thibault

Corresponding Author(s): Charlotte PRONIER, Hopital Pontchaillou

Review Timeline:

Submission Date:	September 18, 2024
Editorial Decision:	October 26, 2024
Revision Received:	December 18, 2024
Accepted:	December 20, 2024

Editor: Jérôme Le Goff

Reviewer(s): Disclosure of reviewer identity is with reference to reviewer comments included in decision letter(s). The following individuals involved in review of your submission have agreed to reveal their identity: Ludmila Perelygina (Reviewer #1)

Transaction Report:

DOI: <https://doi.org/10.1128/spectrum.02348-24>

Re: Spectrum02348-24 (Granuloma and persistent detection of wild-type rubella virus in an immunocompromised patient.)

Dear Dr. Charlotte Pronier PRONIER:

Thank you for the privilege of reviewing your work. Below you will find my comments, instructions from the Spectrum editorial office, and the reviewer comments.

The full phylogenetic tree with the 32 WHO reference viruses should be included to support classification of the virus as genotype 2B.

Revision Guidelines

Sincerely,
Jérôme Le Goff
Editor
Microbiology Spectrum

Reviewer #1 (Comments for the Author):

Charlotte Pronier and colleagues present a case report describing rubella virus-associated granuloma in the lymph node of a 29-year-old immunocompromised, vaccinated patient. While there is a report in the literature of vaccine-derived RuV persistent shedding into respiratory secretions, CSF, and urine, this case highlights persistent shedding of wild type virus into patient's

plasma over 25-month period, which is a novel finding. Detection of wild type virus in granuloma of a fully vaccination person challenges a previous assumption that RuV persisting in granulomas of vaccinated individuals are vaccine-derived, and therefore sequence confirmation is essential for proper diagnosis. Given ongoing efforts towards rubella elimination and proposed rubella eradication, this finding has important implications for rubella surveillance and infection control.

The writing is clear and concise, and the figures are of good quality. The article will be of interest to a broad audience, including dermatologists, virologists, immunologists, and public health specialists.

To further improve the manuscript, please consider the following comments

1. Consider including a full phylogenetic tree with the 32 WHO reference viruses as panel A in Figure 1 and presenting the 2B subtree as panel B. It will provide better evidence for RuV classification as 2B. To further enhance the analysis, you might consider reaching out any of Regional or Global Rubella Reference Labs to obtain sequences of the 25 reference strains representing 7 lineages within 2B genotype. This could help more accurately identify the source of the virus. Use of lineage reference strains to improve rubella surveillance has been demonstrated in several publication (e.g., <https://doi.org/10.1016/j.vaccine.2022.09.084>; <https://doi.org/10.1099/jgv.0.000680>)
2. Consider submitting the RuV sequence to GenBank and including a GB accession number in the report. This will enable other investigators in the rubella community to use this sequence for comparative analyses.
3. Line 73. To-date, 4 cases of wt RuV within granulomas have been reported, not 3. The 4th case involves wt RuV of the 1a genotype, not derived from the vaccine virus strains (<https://doi.org/10.1002/jvc2.268>).
4. Line 76. Wild type RuVs in these 2 published cases were not classified as genotype 2C as they differ from 2C by over 10%. It was suggested that these 2 viruses represent a distinct genotype missed by rubella surveillance, which is far from being optimal.
5. Presence of RuV and RuV-specific intrathecal antibody suggests ongoing CNS infection and inflammation. Have any other viruses been detected in this patient's CSF, such as CMV or VZV? Did this patient have any neurological symptoms?
6. According to Figure 1, the RuV sequences differed across the locations and time points. Authors should mention this in the text and provide an explanation for observed variations.
7. Have the authors attempted to culture the viruses?
8. I suggest emphasizing the impact that long term shedding of wild type RuV may have on rubella elimination/eradication efforts.
9. Line 44. Minor comment - please rephrase "rubella testing was tested".

Reviewer #1 (Comments for the Author):

1. Consider including a full phylogenetic tree with the 32 WHO reference viruses as panel A in Figure 1 and presenting the 2B subtree as panel B. It will provide better evidence for RuV classification as 2B. To further enhance the analysis, you might consider reaching out any of Regional or Global Rubella Reference Labs to obtain sequences of the 25 reference strains representing 7 lineages within 2B genotype. This could help more accurately identify the source of the virus. Use of lineage reference strains to improve rubella surveillance has been demonstrated in several publication (e.g., <https://doi.org/10.1016/j.vaccine.2022.09.084>; <https://doi.org/10.1099/jgv.0.000680>)

- Reply from authors: We thank the reviewer for the helpful comments. As suggested, we now show the tree with all WHO reference sequences and the additional lineage reference strains. Given the limited size of the tree, we think the 2B subtree is actually not needed and prefer just to show the full tree as Figure 1.

2. Consider submitting the RuV sequence to GenBank and including a GB accession number in the report. This will enable other investigators in the rubella community to use this sequence for comparative analyses.

- Reply from authors: We have submitted the sequences to GenBank and are currently waiting for the accession numbers.

3. Line 73. To-date, 4 cases of wt RuV within granulomas have been reported, not 3. The 4th case involves wt RuV of the 1a genotype, not derived from the vaccine virus strains (<https://doi.org/10.1002/jvc2.268>).

- Reply from authors: We thank the reviewer for bringing this to our attention. We have changed the number accordingly, added the reference and included a short description of this case in the manuscript: “The fourth case was a 70-year-old man with unknown vaccine status. Cutaneous lesions of the left leg slowly evolved for over 30 years in the absence of immunodeficiency. Wild-type RuV genotype 1a was identified from the skin granuloma during his care for lymphocytic lymphoma” (lines 92-95)

4. Line 76. Wild type RuVs in these 2 published cases were not classified as genotype 2C as they differ from 2C by over 10%. It was suggested that these 2 viruses represent a distinct genotype missed by rubella surveillance, which is far from being optimal.

- Reply from authors: Thank you for pointing this mistake out. We have rephrased the text as follows: “In both cases, a distinct genotype missed by rubella surveillance was amplified from skin biopsies (multiple body sites for the first patient and the left arm for the second).” (lines 88-90)

5. Presence of RuV and RuV-specific intrathecal antibody suggests ongoing CNS infection and inflammation. Have any other viruses been detected in this patient's CSF, such as CMV or VZV? Did this patient have any neurological symptoms?

- Reply from authors: Thank you for highlighting this important point about the associated neurological signs. Indeed, the patient presented neurological signs that led to lumbar punctures being performed at different times (M9 and M13). We added the clinical elements we have in the text as follows (lines 59-69). "In December 2020, the patient was admitted to hospital in the presence of neurological symptoms consistent with sensorimotor neuropathy of the lower limbs. These symptoms evolved and led to total hearing loss and tetraparesis in 2022. In this context, two lumbar punctures have been performed at different times from the first viral detection in the lymph node, at M9 (when the neurological signs started) and at M13 (when the motor deficits worsened), respectively. At M13, cerebral MRI showed discrete impairment of cerebral trophicity and hypersignals in the deep white matter, particularly in bilateral parietal periventricular areas, without clear aetiological orientation. CSF analyses revealed the presence of RuV RNA at M9 and M13 and specific intrathecal anti-RuV IgG synthesis at M9 (Table 1) while Herpes Simplex Virus (HSV) and Varicella-Zoster Virus (VZV) intrathecal syntheses were negative." This is now specified in the text. The patient was seronegative for CMV.

6. According to Figure 1, the RuV sequences differed across the locations and time points. Authors should mention this in the text and provide an explanation for observed variations.

- Reply from authors: We have added the following text to the article: (lines 100-110) "Interestingly, the sequences of our case obtained at different time points and from different specimen types and locations varied by up to 22 nucleotides within the 739 nucleotide region analyzed. As reported previously for a skin biopsy and a nasopharyngeal swab from a granuloma patient with primary immune deficiency and immunodeficiency-related vaccine-derived rubella viruses from Louisiana (ref Perelgyna 2019), we obtained different sequences derived from different clinical specimens collected at the same time (Plasma 1 and CSF 2, Figure 1). In addition, the sequences from the same specimen type differed between the different collection time points and interestingly also between two plasma samples collected at the same time point (Plasma 2 and 3, Figure 1). These findings corroborate the hypothesis of continuous and independent virus evolution at different body sites and a complex virus population structure in chronically infected patients (ref Perelgyna 2019), even if infected with a wild-type-derived rubella virus."

7. Have the authors attempted to culture the viruses?

- Reply from authors: No, we did not attempt virus culture because most of the specimen types collected are not recommended for virus isolation in cell culture. After conducting the different analyses, there was not enough material left of PCR positive sample(s) to attempt virus isolation.

8. I suggest emphasizing the impact that long term shedding of wild type RuV may have on rubella elimination/eradication efforts.

- Reply from authors: We thank the reviewer for this suggestion and have added the following text to the manuscript: “The impact of long-term shedding of wild-type RuV on rubella eradication efforts is currently difficult to estimate. Positive cultures of both VDRV and wild-type RuV from granuloma raise the question of the transmissibility of these viruses and thus the risk of this virus reservoir for elimination efforts.” (lines 152-156)

9. Line 44. Minor comment - please rephrase "rubella testing was tested".

- Reply from authors: As suggested, the sentence has been rephrased: “Given these initial negative results for the infectious agents classically found in granulomas, rubella testing was conducted.” (lines 47-48)

Re: Spectrum02348-24R1 (Granuloma and persistent detection of wild-type rubella virus in an immunocompromised patient.)

Dear Dr. Charlotte Pronier PRONIER:

Your manuscript has been accepted, and I am forwarding it to the ASM production staff for publication. Your paper will first be checked to make sure all elements meet the technical requirements. ASM staff will contact you if anything needs to be revised before copyediting and production can begin. Otherwise, you will be notified when your proofs are ready to be viewed.

Sincerely,
Jérôme Le Goff
Editor
Microbiology Spectrum